# Changes in the Relationship Between Gray Matter, Functional Parameters, and Quality of Life in Patients with a Post-Stroke Spastic Upper Limb After Single-Event Multilevel Surgery: Six-Month Results from a Randomized Trial

**DOI:** 10.3390/diagnostics15081020

**Published:** 2025-04-16

**Authors:** Patricia Hurtado-Olmo, Pedro Hernández-Cortés, Ángela González-Santos, Lourdes Zuñiga-Gómez, Laura Del Olmo-Iruela, Andrés Catena

**Affiliations:** 1Hand & Upper Limb Surgery Unit, Orthopedic Surgery Department, San Cecilio University Hospital of Granada and Spain, 18016 Granada, Spain; phurtadoolmo@gmail.com; 2Surgery Department, School of Medicine, Granada University, 18016 Granada, Spain; 3Instituto de Investigación Biosanitaria IBS, 18012 Granada, Spain; 4Department of Physical Therapy, Faculty of Health Science, University of Granada, 18071 Granada, Spain; angelagonzalez@go.ugr.es; 5BIO277 Group, A02-Cuídate, Instituto de Investigación Biosanitaria, 18012 Granada, Spain; 6Rehabilitation Department, San Cecilio University Hospital of Granada, 18007 Granada, Spain; lzunigagomez@hotmail.com (L.Z.-G.); olmoiruela@gmail.com (L.D.O.-I.); 7Faculty of Psychology, University of Granada, 18011 Granada, Spain; acatena@go.ugr.es

**Keywords:** stroke, spastic upper limb, surgical procedures, gray matter volume, magnetic resonance imaging, neuronal plasticity

## Abstract

**Introduction:** Advanced magnetic resonance imaging (MRI) techniques in neuroplasticity evaluations provide important information on stroke disease and the underlying mechanisms of neuronal recovery. It has been observed that gray matter density or volume in brain regions closely related to motor function can be a valuable indicator of the response to treatment. **Objective**: To compare structural MRI-evaluated gray matter volume changes in patients with post-stroke upper limb spasticity for >1 year between those undergoing surgery and those treated with botulinum toxin A (BoNT-A) and to relate these findings to upper limb function and quality of life outcomes. **Materials and Methods**: *Design.* A two-arm controlled and randomized clinical trial in patients with post-stroke upper limb spasticity. *Participants*. Thirty post-stroke patients with spastic upper limbs. *Intervention.* Participants were randomly assigned (1:1 allocation ratio) for surgery (experimental group) or treatment with BoNT-A (control group). *Main outcome measures*. The functional parameters were analyzed with Fugl-Meyer, Zancolli, Keenan, House, Ashworth, pain visual analogue, and hospital anxiety and depression scales. Quality of life was evaluated using SF-36 and Newcastle stroke-specific quality of life scales. The carer burden questionnaire was also applied. Clinical examinations and MRI scans were performed at baseline and at six months post-intervention. Correlations between brain volume/thickness and predictors of interest were examined across evaluations and groups. **Results**: Five patients were excluded due to the presence of intracranial implants. Eleven patients were excluded from analyses since they were late dropouts. Changes were observed in the experimental group but not in the control group. Between baseline and six months, gray matter volume was augmented at the hippocampus and gyrus rectus and cortical thickness was increased at the frontal pole, occipital gyrus, and insular cortex, indicating anatomical changes in key areas related to motor and behavioral adaptation These changes were significantly related to subjective pain, Ashworth spasticity scale, and Newcastle quality of life scores, and marginally related to the carer burden score. **Conclusions**: The structural analysis of gray matter by MRI revealed differences in patients with post-stroke sequelae undergoing different therapies. Gray matter volume and cortical thickness measurements showed significant improvements in the surgery group but not in the BoNT-A group. Volume was increased in areas associated with motor and sensory functions, suggesting a neuroprotective or regenerative effect of upper limb surgery.

## 1. Introduction

Stroke is the principal cause of permanent disability in adults [1]. More than one-third of post-stroke patients develop spasticity, which requires lifelong medical treatment and reduces independence in daily life activities [2,3]. Stroke survivors are commonly dissatisfied with the recovery of upper limbs, even when only mildly affected, due to the negative effect on their quality of life [4].

The adult brain retains plasticity and functional reorganization capacities throughout life [5], and stroke patients generally experience a degree of spontaneous improvement in the subacute phase due to neuroplasticity, which can be modified by appropriate therapies [6]. Mechanisms involved in this process include synaptic plasticity [7], axonal sprouting/connection regeneration [8], neurogenesis [9], cortical functional reorganization [10], and functional network modification [11].

The study of neuroplasticity by advanced magnetic resonance imaging (MRI) techniques yields important information on stroke and neuronal recovery mechanisms. In this way, the density or volume of gray matter in brain regions related to motor function has proved to be a valuable indicator of the response to treatment [6].

The first-choice treatment for localized spasticity is currently the intramuscular injection of botulinum toxin A (BoNT-A) [12,13,14]. Besides the peripheral effects of this therapy, functional MRI (fMRI) scans have revealed brain modulation in patients treated with BoNT-A after a stroke [15,16,17] or traumatic brain injury [18]. When used to treat focal spasticity, BoNT-A was found to increase the activation of brain areas involved in motor control and to improve cerebellar connectivity related to motor impairment [19,20]. However, to the best of our knowledge, there has been no investigation into the effect of BoNT-A treatment on gray matter volume in post-stroke patients.

Various surgical procedures have been used to improve the function, hygiene, and esthetics of spastic upper limbs and to reduce pain [21]. Available approaches include single-event multilevel surgery, which combines soft tissue releases and elongations with hyponeurotization, tendon transfers, and joint stabilization procedures [22]. No structural MRI or fMRI study has been published on brain changes after spastic hand/upper limb surgery.

The objective of this study was to use structural MRI to compare 6-month changes in gray matter volume at >12 months post-stroke between patients with upper limb spasticity treated with BoNT-A and those undergoing surgery and to relate these findings to upper limb function and quality of life outcomes. It was hypothesized that structural changes to the brain might be greater with surgery because it produces permanent changes in upper limb spasticity and position.

## 2. Material and Methods

### 2.1. Design

A two-arm randomized controlled clinical trial was conducted in patients with post-stroke upper limb spasticity, randomly assigning participants (1:1) to the control group for BoNT-A treatment or to the experimental group for surgery. All participants underwent a brain study via structural MRI at baseline and six months later. This trial project (Ref: PI20/01574) was funded by the Instituto de Salud Carlos III (ISCIII) and co-funded by the European Union. The trial was registered at Clinical Trials.gov with registration number NCT06392633 on 30 April 2024.

The flowchart of the study is shown in Figure 1.

### 2.2. Patients

The sample size was estimated for 5% type I error probability, 80% power, and a large effect size difference (0.8 standard deviations). The study inclusion criteria were age >18 years, post-stroke upper limb spasticity, post-stroke interval of >12 months, agreement to surgery, and the signing of informed consent for study participation (by patient or legal representative). The exclusion criteria were American Society of Anesthesiologists (ASA) grade ≥IV for operative risk, involuntary (extrapyramidal) movements, inability to correctly complete the questionnaire, surgically untreatable deformities, and impossibility of follow-up for ≥12 months. Recruitment began on 9 January 2023. The trial was not interrupted for any reason.

The recruited patients were clinically assessed by a rehabilitation specialist at Clinic San Cecilio University Hospital of Granada (CSCUH) using a neurological examination and brief standardized medical history interview. Participants meeting the eligibility criteria underwent MRI examination and were subsequently assessed by a hand and upper limb surgeon, who examined the shoulder, elbow, wrist, and hand under peripheral nerve block anesthesia. Each participant then received an individualized proposal for single-session surgery that combined soft tissue procedures (tenotomy, tendon elongation, or tendon transfer), selective neurectomies, and bone procedures (osteotomies and arthrodesis).

The random assignment of participants to study groups was conducted after the enrolment of all participants (*n* = 30) to reduce the risk of selection bias. Randomization was performed by an independent member of the research team who was not involved in the data collection, using an allocation sequence based on a computer-generated random number.

Figure 2 depicts MRI images of the participants at baseline. The lesion localization, stroke severity, and time since stroke were equivalent between the two study groups.

### 2.3. Experimental Group

The 15 patients in this group were placed on the surgery waiting list of the CSCUH and underwent an anesthetic risk assessment by the Anesthesiology Department. The mean interval between stroke onset and study enrolment was 8.09 years. They were admitted to hospital the day before surgery, having replaced antiaggregant/anticoagulant therapy with low-molecular-weight heparin. They were anesthetized in accordance with their preanesthetic study results for the single-event multilevel surgery, which involved the shoulder (tenotomy of *pectoralis major* and/or subscapularis elongation), elbow (biceps and brachialis elongation or musculocutaneous nerve hyponeurotization), forearm (pronator teres tenotomy or Scaglietti procedure [flexor-pronator mass release]), wrist (arthrodesis, tendon elongation, tendon transfers, especially superficialis-to-profundus transfers), and hand (tendon elongation, ulnar nerve motor branch neurectomies, and arthrodesis). All 15 patients in this group underwent the first MRI, but only 8 agreed to the second MRI at six months.

### 2.4. Control Group

The 15 patients in this group received injections of BoNT-A in spastic upper limb muscles every four months in the hospital rehabilitation department. The mean interval between stroke onset and study enrolment was 7.52 years. Five patients were excluded due to the presence of metallic intracranial implants. Out of the remaining ten patients, four refused the second MRI at six months, and both MRI scans were available for only six patients.

### 2.5. Data Collection

At baseline and at six months post-enrolment, limb function and quality of life were evaluated by a rehabilitation specialist with >25 years’ experience during a single session at the Surgery Department of the School of Medicine of the University of Granada, and MRI scans were performed at the Mind, Brain, and Behavior Research Center of the university.

### 2.6. Clinical Evaluation

At baseline, data were gathered from all participants regarding age, sex, type of stroke, date of stroke, interval between stroke and treatment, Functional Ambulation Categories (FAC) score for gait [23], Folstein mini-mental scale score for cognitive status [24], Barthel Index for functional independence [25], and ASA grade for anesthetic risk [26]. Upper limb lesions were classified as functional or hygienic and elbow, wrist, and hand deformities were categorized according to Keenan et al. [27]

#### 2.6.1. Evaluation of Upper Limb Functionality

An evaluation of upper limb functionality was conducted at baseline and six months later using the following methods:Modified Ashworth scale for spasticity [28]. The score ranges from 0 to 5. The result is positive when significant improvement is achieved in at least one of the five sites evaluated (elbow, forearm, wrist, thumb, fingers).Nine-point hand function scale by House [29].Fugl-Meyer post-stroke recovery scale for shoulder, elbow, wrist, and hand domains with 33 items scored on a three-point scale (0–2), with a total score ranging from 0 to 66 [30].

#### 2.6.2. Evaluation of Hygienic/Nonfunctional Outcomes

This evaluation was conducted at baseline and at six months later using the following methods:Goal Attainment Scaling [31] for hygiene, esthetics, and pain, using a five-point scale (−2 to +2).Carer burden score [22] for nail cutting, hand palm cleaning, underarm cleaning, and arm dressing), using a five-point Likert-type scale.Pain Visual Analogue Scale (VAS) [32].

#### 2.6.3. Evaluation of Quality of Life

This was conducted at baseline and six months using the following methods:Spanish-validated version of SF-36 [33].Newcastle Stroke-Specific Quality of Life Measure [34].

#### 2.6.4. Evaluation of Quality of Sleep, Anxiety, and Depression

This was conducted at baseline and six months using the following methods:SATED sleep health questionnaire, containing five items scored from 0 to 2 [35].Spanish-validated version of the Hospital Anxiety and Depression Scale (HADS questionnaire) [36], containing 14 items scored from 0 to 3 (higher score = increased severity).

### 2.7. MRI Assessment and Segmentation

MRI scans were taken at baseline and six months with a Siemens 3T Prisma MRI scanner. Three-dimensional volumes were obtained using a T1-weighted MPRAGE sequence in sagittal orientation with 1 mm isotropic resolution slides, FOV = matrix 256 × 256 × 208, with TR (TE/TI of 2300/2.98/900, flip angle 9°). The fully automated volBrain segmentation pipeline (https://volbrain.net/ (accessed on 10 October 2024)) was used to measure gray matter volume and thickness [37]. This pipeline requires no training or setup and has demonstrated similar results to those obtained by expert manual segmentation [38,39]. volBrain uses a Spatially Adaptive Non-Local Means (SANLM) Filter to reduce spatial noise [40].

There were no changes to trial outcomes after the trial commenced.

### 2.8. Statistical Analysis

The R mice package version 3.17.0 was used to impute missing values by the predictive mean matching method (pmm) and default m (=5) parameter with 50 iterations, with all other numerical variables as predictors. A correlation analysis of the imputed results was performed to detect differences in brain volume and thickness measurements between pre- and post-treatment in relation to the scale scores of interest (Ashworth, House, Keenan, Fugl-Meyer, Career Burden, Newcastle, and SF). After computing differences in the variables of interest between the evaluations for each study group, Fisher’s test was applied to compare correlations between the groups, considering false discovery ratio (FDR)-corrected *p*-values. In this way, correlations between brain volume/thickness and predictors of interest were examined across evaluations and groups.

## 3. Results

The study population comprised 30 patients. Table 1 displays their demographic characteristics and descriptive statistics for the variables of interest, showing differences in evaluations between baseline and six months for each group. Given the small number of participants in the groups, a non-parametric approach was adopted for the comparisons. The rate of change between evaluations was computed as the difference in score between after and before treatment divided by the pre-treatment score. A positive rate of change therefore signifies a decrease in the score.

No significant differences were observed between before and after treatment in the control group, whereas the experimental (surgery) group showed a significant difference between evaluations in the pain VAS score (*p* < 0.009) and Newcastle measure (*p* < 0.05), with a close-to-significant difference in carer burden score (*p* = 0.093). Table 2 exhibits the mean rates of change with a non-parametric p-level for between-group comparisons; these comparisons only considered the ten participants who underwent both MRI studies (six from the control group and nine from experimental group).

The experimental group showed significantly higher rates of change in the Ashworth spasticity scale and pain VAS scores and close-to-significantly higher rates of change in House scale and carer burden scores.

The brain change index (BCI) was computed as the difference in volume and cortical thickness values between post- and pre-treatment divided by the pre-treatment values. Significant between-group differences in BCI are displayed in Table 3a for brain volumes and in Table 3b for cortical thicknesses, along with the corresponding *p*-values (non-parametric Mann–Whitney test).

After computing differences in study variables between pre- and post-treatment, correlations between differential variables in each group were analyzed using the Fisher z-test to determine the significance of differences in between-group correlations. Table 4 exhibits the results for brain volume and Table 5 those for cortical thickness. Gray matter volume was increased in the hippocampus and gyrus rectus and cortical thickness was increased at the frontal pole, occipital gyrus, and insular cortex between baseline and six months, indicating superior functional connectivity in key areas related to motor and behavioral adaptation. These changes were significantly related to VAS, Ashworth spasticity scale, and Newcastle quality of life scores, and marginally related to the carer burden score.

## 4. Discussion

The effects of a stroke are influenced by multiple factors, including the extension and size of the infarction and the changes in neuroplasticity [41,42,43].

Upper limb motor dysfunction is one of the most frequent post-stroke sequelae and the main cause of functional and quality of life impairments [44]. The spastic upper limb can be treated by periodical BoNT-A injections [12,13,14], robot-assisted rehabilitation techniques [45,46], transcranial magnetic stimulation [47], extracorporeal shock wave therapy [48,49], neurofeedback [50], and training with a brain–machine interface [51], among others. A surgical approach has generally been displaced by BoNT-A and is now selected in only a small number of patients [52].

The emergence of advanced MRI techniques offers a new perspective on post-stroke neuroplasticity and the response to treatment in terms of activation patterns, functional connectivity, and structural changes [53]. The investigation of post-stroke brain changes has largely focused on variations in brain function [54,55], and there has been little research on changes in cortical thickness or in gray matter volume [56], whose modification is considered one of the most important consequences of neuroplasticity [57].

Questions have been raised about the effectiveness of certain treatments [58] due to the lack of robust recovery markers and the differences in evaluation criteria among studies, which hamper their comparison [53]. MRI studies of structural and functional brain changes can be useful to assess the effectiveness of treatments, and researchers have used brain MRI scans to describe functional changes in patients treated with BoNT-A [15,16,17], but not to examine functional or structural changes in patients treated with surgery.

To the best of our knowledge, this is the first study to use structural MRI to evaluate the gray matter of surgically treated patients with stroke sequelae in upper limb. Significant changes in gray matter volume and cortical thickness were observed at six months in the surgically treated group but not in those receiving BoNT-A injections. These changes were significantly associated with improvements in functional outcomes, pain relief, spasticity, patient-perceived quality of life, and carer burden.

In the surgery group, volumes were not only increased in the cerebellum, known to be involved in motion and sensorimotor integration [59], but also at temporal (e.g., hippocampus), frontal (e.g., gyrus rectus and middle superior frontal), and posterior (e.g., cuneus and occipital fusiform) sites. Hence, the surgical procedure appears to have a positive effect on numerous brain areas involved in the senso-motor network. Given the significant role played by the hippocampus and prefrontal cortex in different cognitive and emotional processes, these findings indicate a possible systemic involvement of the frontolimbic network. This could result in post-surgical emotional changes [60,61] that are likely independent of the localization of the lesion.

The time after ischemic stroke onset can be categorized as the acute (first seven days), subacute (one week to six months), or chronic (more than six months) phase [43]. The clinical recovery of intrinsic neural mechanisms can take place soon after stroke onset, including synaptic changes, the restoration of neuronal excitability and network responsiveness to the stroke, and tissue repair processes, and the brain is most primed for recovery during the first few weeks. A plateau is then reached between three and six months, and spontaneous recovery is not usually observed beyond six months, resulting in chronic deficits [62].

Likewise, the cortical thickness was significantly increased after the surgery in multiple brain areas, including frontal (e.g., frontal pole, anterior/posterior orbital gyrus), temporal (e.g., planum polare and inferior temporal), and occipital (e.g., calcarine, lingual, and inferior, middle, and superior occipital gyrus) sites.

Functional and quality of life outcomes were more strongly associated with cortical thickness than brain volume changes. In the case of the Ashworth spasticity scale, the change in score between before and after surgery was negatively correlated with the volume of the right triangular frontal gyrus (TrlFG), a part of the frontal cortex implicated in semantic processing [63]. A strong negative association was observed between the Ashworth scale score and the thickness of the sensorimotor cortex in the BonT-A group but not in the surgery group, and this signifies a close relationship between lower cortical thickness at this site and greater spasticity [64]. The same negative association was observed for total temporal thickness (i.e., occipital left and total insular cortex). Morphological changes have been described in degenerative motor diseases such as lateral amyotrophic sclerosis [65], and similar findings have been reported for the insular [66] and occipital cortex [67].

House and Fugl-Meyer scale scores have previously been related to both post-stroke changes in temporal and occipital brain areas, including reduced cortical thickness [68], as well as partial post-intervention brain recovery [69]. The weaker correlation observed for the Fugl-Meyer score may be explained by an indirect effect or by the multidimensional nature of this scale, which considers not only motor function but also pain, passive mobility, and sensory aspects related to spasticity.

Associations with the carer burden score significantly differed between the present groups in relation to the left posterior orbital and postcentral gyrus asymmetry. In the surgery group, a negative correlation was observed between posterior orbital cortical thickness and carer burden (increased thickness = greater burden). Interestingly, this part of the cortex has been linked not only to novelty detection and memory but also to motivational control and goal-directed behaviors [70]. These are crucial for adaptation to daily life, and a reduction in this cortex has been associated with obsessive compulsive disorders, among others [71].

Finally, the changes in volume and cortical thickness of the middle occipital gyrus observed in the present study have also been described in patients with chronic pain [71,72].

Limitations. This study is limited by the reduced sample size and short follow-up period. We found asymmetry in the inclusion of patients with metallic intracranial implants between the two groups and a high number of dropouts, but we believe that this is compensated by the statistical treatment carried out as demonstrated by the significant differences observed. Although the lesion localization, stroke severity, and time since stroke were equally distributed between the two study groups, the rates of cortical and subcortical strokes were not compared. In addition, patients who underwent surgery no longer required toxin treatment, making it impossible to differentiate between changes produced by the surgery and those due to the interruption of the toxin treatment. The strengths of the study include its randomized design and the novelty of its relevant findings.

Perspectives. Knowledge of the pathophysiology of stroke and the biology of gray matter reorganization could support the development of therapies to enhance recovery after acute central nervous system injury. Further research is needed on changes in white matter and functional MRI in patients with chronic stroke to evaluate surgery and other upper limb rehabilitation therapies. In particular, structural and functional brain MRI scans can help to determine the relevance of the novel approach to spasticity surgery using nerve transfer procedures [73].

## 5. Conclusions

The structural analysis of gray matter by MRI can reveal differences in patients with post-stroke sequelae undergoing different therapies.

Gray matter volume and cortical thickness measurements showed significant changes in the surgery group but not in the BoNT-A group. Volume was increased in areas associated with motor and sensory functions, suggesting a neuroprotective or regenerative effect of upper limb surgery.

## Figures and Tables

**Figure 1 diagnostics-15-01020-f001:**
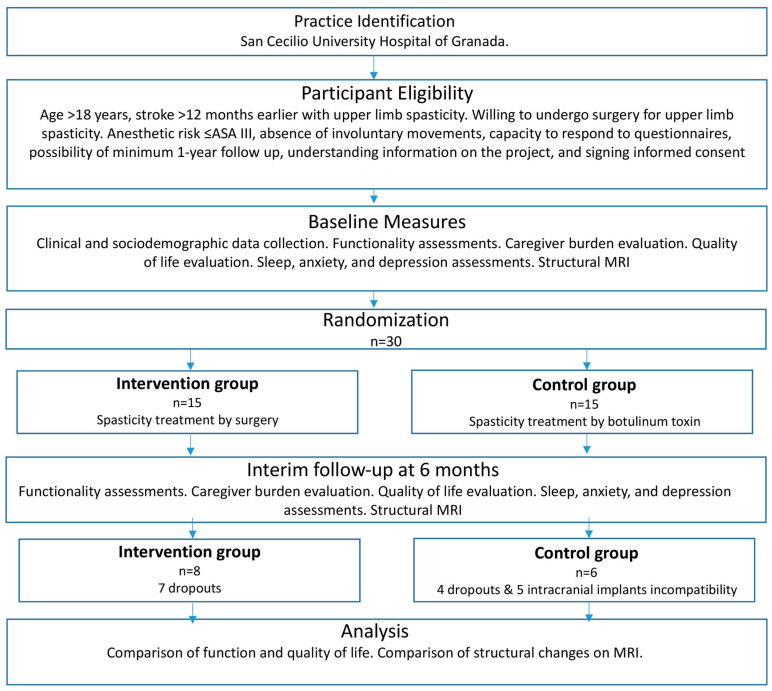
CONSORT diagram.

**Figure 2 diagnostics-15-01020-f002:**
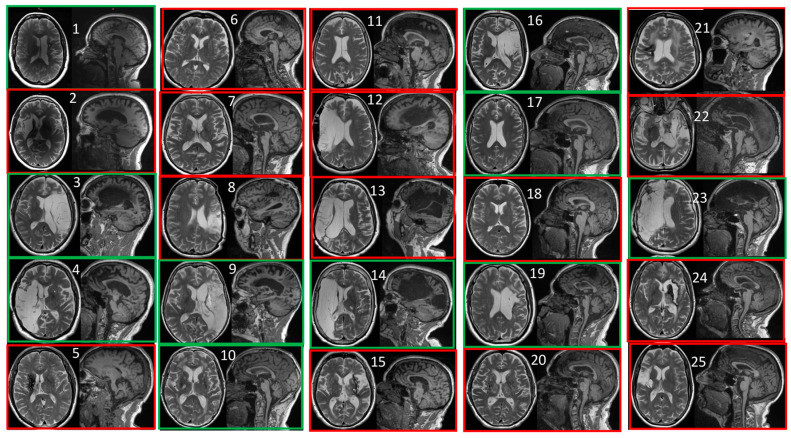
Structural MRI images of each of the patients at baseline. Images corresponding to the botulinum toxin-treated control group are framed in green and those corresponding to the surgically treated experimental group are framed in red. The numbers refer to the order of patient recruitment in the study.

**Table 1 diagnostics-15-01020-t001:** Descriptive statistics, means with standard errors, and non-parametric significance of comparisons of the rate of change between MRI evaluations in each group.

	Baseline		Six Months		Rate of Change	
Variable	Control	Experimental	Control	Experimental	Control	Experimental
*n*	10	15	6	8		
Male sex	7	6	4	5		
Ashworth	3.9 (±0.38)	4.93 (±0.23)	3.83 (±0.60)	2.63 (±0.38)	−0.02	−0.47
House	1.70 (±0.60)	1.67 (±0.41)	1.67 (±0.67)	2.63 (±0.68)	−0.02	0.57
Keenan	4.10 (±0.53)	4.33 (±0.36)	2.83 (±0.83)	1.63 (±0.50)	−0.31	−0.62
Fugl Mayer	44.9 (±9.28)	49.6 (±6.47)	44.5 (±12.3)	62.75 (±8.18)	−0.01	0.27
Carer Burden	6.50 (±1.56)	7.40 (±0.76)	5.33 (±1.96)	1.50 (±1.22)	−0.18	−0.80+
Pain Evaluation	4.70 (±0.58)	4.87 (±1.01)	5.33 (±0.56)	1.75 (±0.75)	0.13	−0.64 **
Newcastle	8.50 (±0.83)	7.51 (±0.69)	8.85 (±1.2)	5.95 (±0.60)	0.04	−0.21 *
SF36	51.5 (±7.59)	52.77 (±5.73)	46.02 (±11)	51.28 (±7.79)	−0.11	−0.03
Age	58.8 (±2.99)	57.93 (±2.88)	58.5 (±4.08)	58.63 (±4.82)	−0.01	0.01

Pain evaluation ** (*p* < 0.009), Newcastle * (*p* < 0.05), carer burden + (*p* = 0.093). Ashworth: Modified spasticity scale; House: Motor hand capability scale of House; Keenan: Spastic hand appearance scale of Keenan; Fugl-Meyer: Upper limb motor function scale; Carer Burden: carer burden scale; Pain evaluation: Pain visual analogue scale; Newcastle: Newcastle stroke-specific quality of life measure; SF36: SF36 health questionnaire scale.

**Table 2 diagnostics-15-01020-t002:** Mean rate of change values and non-parametric significance for the between-group comparison at six months.

	Ashworth	House	Keenan	Fugl Mayer	Carer Burden	Pain Evaluation	Newcastle	SF-36
Control	−0.058	−0.048	−0.267	0.136	−0.048	−0.056	−0.026	0.025
Experimental	−0.460	0.750	−0.575	0.526	−0.852	−0.323	−0.180	0.034
*p*-value	0.002	0.080	0.160	0.360	0.080	0.050	0.230	0.100

Ashworth: Modified Ashworth spasticity scale; House: Motor hand capability scale of House; Keenan: Spastic hand appearance scale of Keenan; Fugl-Meyer: Upper limb motor function scale; Carer Burden: carer burden scale; Pain evaluation: Pain visual analogue scale; Newcastle: Newcastle stroke-specific quality of life measure; SF36: SF36 health questionnaire scale.

**Table 3 diagnostics-15-01020-t003:** (**a**). Significant between-group differences in brain change index (BCI) for brain volumes. (**b**) Significant between-group differences in brain change index (BCI) for cortical thickness.

(a)			
Volumes (cm^3^)	Control	Experimental	*p*-Value
Cerebellum left	−0.032	0.065	0.020
Cerebellum WM total	−0.017	0.114	0.010
Hippocampus total	−0.032	0.177	0.020
GRe total	−0.052	0.237	0.010
GRe left	−0.083	0.212	0.010
MSFG right	−0.021	0.465	0.020
STG total	−0.015	0.789	0.020
STG right	−0.020	0.463	0.010
Cun left	−0.021	0.504	0.020
OFuG left	−0.017	0.273	0.020
**(b)**			
**Thickness (mm)**	**Control**	**Experimental**	***p*-Value**
FRP right	−0.055	0.354	0.005
AOrG total	−0.076	0.032	0.008
POrG right	−0.041	0.232	0.008
PP asym	−0.067	0.137	0.020
ITG asym	−0.067	0.212	0.008
Calc total	−0.074	0.286	0.003
Calc left	−0.149	4.844	0.008
Calc asym	−0.074	0.142	0.020
LiG asym	−0.034	0.113	0.020
OFuG right	−0.037	0.121	0.005
IOG total	−0.036	0.304	0.020
MOG right	−0.038	0.242	0.008
MOG asym	−0.055	0.077	0.013
SOG right	−0.050	0.108	0.003

Note. GRe. Gyrus rectus, MSFG: Middle superior frontal gyrus, STG: Superior temporal gyrus, Cu: Cuneus, OFuG: Occipital fusiform gyrus, FRP: Frontal pole, AOrG: Anterior orbital gyrus, POrG: Posterior orbital gyrus, PP: Planum polare, ITG: Inferior temporal, Calc: Calcarine, LiG: Lingual, OFuG: Occipital fusiform, IOG: Inferior occipital, MOG: Middle occipital, SOG: Superior occipital.

**Table 4 diagnostics-15-01020-t004:** FDR-corrected significant changes in correlations of functional and quality of life measures with brain volumetric values between experimental and control groups across pre-post-treatment sessions. Fisher z-scores are reported, displaying correlations for control and experimental groups in parentheses.

Volume (cm^3^)	Keenan	Fugl-Meyer	Newcastle
TrIFG right	2.87 (−0.89/0.57)	-	-
IOG left	-	−3.26 (−0.76/0.88)	-
PIns left	-	−3.25 (−0.32/097)	-
Vermis	-	-	−3.01 (−0.96/0.30)
AnG total	-	-	3.30 (0.97/−0.29)

Note. TrlFG: Triangular frontal, IOG: Inferior occipital, PIns: Posterior insula, AnG: Angular gyrus. Keenan: Spastic hand appearance scale of Keenan; Fugl-Meyer; Newcastle: Newcastle stroke-specific quality of life measure.

**Table 5 diagnostics-15-01020-t005:** FDR-corrected significant changes in correlations of functional and quality of life measures with brain cortical thickness values between experimental and control groups across pre-post-treatment sessions. Fisher z-scores are reported, displaying correlations for control and experimental groups in parentheses.

Thickness (mm)	Ashworth	House	Keenan	Fugl-Meyer	Carer Burden	SF36
SMC total	−2.9 (−0.95/0.27)	-	-	-	-	-
Temporal total	−3.4 (−0.97/0.35)	-	-	-	-	-
PCu left	3.4 (0.90/−0.75)	-	-	-	-	-
Occipital left	−2.8 (−0.88/0.55)	-	-	-	-	-
Insular total	−2.7 (−0.93/0.31)	-	-	-	-	-
OFuG asym	-	−3.2 (−0.98/−0.03)	-	-	-	-
TTG left	-	-	3.5 (0.98/−0.31)	-	-	-
Parietal right	-	-	-	2.7 (0.88/−0.51)	-	-
POrG left	-	-	-	-	3.3 (0.72/−0.90)	−3.62 (−0.78/0.92)
PoG asym	-	-	-	-	2.6 (0.96/0.04)	-
MOG left	-	-	-	-	-	−3.59 (−0.93/0.75)

Note. SMC: Sensorimotor cortex, PCu: Precuneus, OFuG: Occipital fusiform gyrus, TTG: Transverse temporal gyrus, POrG: Posterior orbital, PoG: Postcentral gyrus, MOG: Middle occipital gyrus. Ashworth: Modified Ashworth spasticity scale; House: Motor hand capability scale of House; Keenan: Spastic hand appearance scale of Keenan; Fugl-Meyer: Upper limb motor function scale; Carer Burden: carer burden scale; Pain evaluation: Pain visual analogue scale; SF36: SF36 health questionnaire scale.

## Data Availability

The data presented in this study are available on request from the corresponding author.

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
