# Peer review of "Changes in the Relationship Between Gray Matter, Functional Parameters, and Quality of Life in Patients with a Post-Stroke Spastic Upper Limb After Single-Event Multilevel Surgery: Six-Month Results from a Randomized Trial"

_diagnostics, 2025, doi:10.3390/diagnostics15081020_

Round 1
Reviewer 1 Report
Comments and Suggestions for Authors
This manuscript presents a randomized controlled trial (RCT) comparing single-event multilevel surgery with botulinum toxin A (BoNT-A) injections in post-stroke patients with spastic upper limb. The study evaluates gray matter changes via MRI, functional recovery, pain relief, and quality of life over a six-month period.
While the topic is clinically relevant, and the study offers novel insights into cortical plasticity post-surgery, there are methodological flaws, statistical inconsistencies, and writing deficiencies that need to be addressed. The sample size is small, the dropout rate is high, and MRI findings are not critically analyzed in a mechanistic manner.
Below is a detailed critical review focusing on content, methodology, and statistical analysis.
- The study correlates gray matter changes with clinical improvement, but no causality can be inferred. How do we know that observed cortical thickness/volume increases are direct effects of surgery and not due to normal post-stroke reorganization? The mechanistic link between surgery and neuroplasticity is speculative. Can the authors provide supportive evidence from basic neuroscience?
- The original cohort had 30 patients, but after exclusions and withdrawals, only 8 surgical patients and 6 BoNT-A patients completed the second MRI. A dropout rate exceeding 50% severely impacts statistical power. Was an intention-to-treat (ITT) analysis performed to minimize bias?
- Surgery provides a permanent effect, whereas BoNT-A requires repeated injections. The control group continued BoNT-A injections every 4 months, potentially introducing confounding effects. How do we separate the effects of surgery itself from the discontinuation of BoNT-A?
- Lesion location, stroke severity, and time since stroke significantly impact recovery. Were these factors equally distributed between groups? Did patients with cortical vs. subcortical strokes exhibit different responses?
- Why were pain and quality-of-life measures strongly correlated with gray matter changes, while motor function (Fugl-Meyer) showed weaker correlations? Could this indicate a placebo effect or an indirect mechanism (e.g., pain relief enabling better movement)?
- MRI scans were performed at six months, but neuroplasticity often continues for years post-stroke. Do the authors plan 12-month or 24-month follow-ups to confirm these findings?
- The hippocampus and gyrus rectus are not primary motor regions—why were these structures affected? Could these changes be related to emotional regulation or cognitive adaptation rather than motor function?
- Did surgical technique (e.g., tendon transfers, neurectomies) influence outcomes? Was there heterogeneity in surgical approaches among patients?
- The study claims a large effect size (0.8 SD) for power calculations, but: This is overly optimistic for a neurorehabilitation study. With n = 8 (surgery) vs. n = 6 (BoNT-A), the statistical power is likely too low for detecting moderate effects. Were confidence intervals calculated?
- VolBrain was used for segmentation, but no details are given about quality control. Motion artifacts are common in stroke patients—were scans visually inspected for motion-related errors? Were baseline differences in brain volume accounted for in analysis?
- Given the small sample, non-parametric tests (Mann-Whitney, Fisher’s exact test) were used, but: Multiple comparisons were performed across Ashworth, House, Keenan, Fugl-Meyer, Carer Burden, SF-36, Newcastle QOL, pain VAS, MRI metrics. Was a Bonferroni or False Discovery Rate (FDR) correction applied? Without correction, many significant p-values (e.g., p = 0.02) may be false positives.
Reviewer 2 Report
Comments and Suggestions for Authors
The work is dedicated to MRI-based follow-up patterns in post-stroke patients for different therapeutic options (such as botulotoxin) and grading scales. The quality of material presentation is fine. However, I recommend some minor points to be corrected.
- No perspectives are present at the discussion, and this reduces interest for clinical workers. Please add limitations and perspectives in details in a separate section.
- The abstract is too long, some extra information from M&M is given. Please shorten the abstract's volume.
- Tables are difficult to read in their current format. Please mark different lines by color or intervals.
- Some minor mistakes are found throughout the text; please re-read (no commas at certain sites, extra spaces, etc.)
- Please check the abbreviations, they are sometimes used with no explication, e.g. LZG at Line 156 (over 10 throughout the text).
Author Response
Please, see the attached document. Thank you
